# KL Divergence Optimization with Entropy-Ratio Estimation for Stochastic GFlowNets

## Abstract

This paper introduces a novel approach for optimizing Generative Flow Networks (GFlowNets) in stochastic environments by incorporating KL divergence objectives with entropy-ratio estimation. We leverage the relationship between high and low entropy states, as defined in entropy-regularized Markov Decision Processes (MDPs), to dynamically adjust exploration and exploitation. Detailed proofs and analysis demonstrate the efficacy of this methodology in enhancing mode discovery, state coverage, and policy robustness in complex environments.

## 1 Introduction

Generative Flow Networks (GFlowNets) (Bengio et al., 2021a;b) have recently gained attention for their application in a variety of tasks, such as molecule discovery (Bengio et al., 2021a; Jain et al., 2022b), biological sequence design (Jain et al., 2022a), and robust scheduling (Zhang et al., 2023). GFlowNets learn policies that generate objects $x \in \mathcal{X}$ sequentially, where the generation process is similar to Monte-Carlo Markov chain (MCMC) methods (Metropolis et al., 1953; Hastings, 1970; Andrieu et al., 2003), generative models (Goodfellow et al., 2014; Ho et al., 2020), and amortized variational inference (Kingma & Welling, 2013). This sequential process of generating objects through a policy also closely resembles reinforcement learning (RL) (Sutton & Barto, 2018).

## 2 Background

Generative Flow Networks (GFlowNets) are variational inference algorithms designed to treat sampling from a target probability distribution as a sequential decision-making process (Bengio et al. (2021a;b)). Below, we briefly summarize the formulation and primary training algorithms for GFlowNets. Consider a fully observed, deterministic Markov Decision Process (MDP) with a state space $\mathcal{S}$ and a set of actions $\mathcal{A} \subseteq \mathcal{S} \times \mathcal{S}$. The MDP has a designated *initial state* $\mathbf{s}_0$, and certain states, called *terminal states*, are designated as having no outgoing actions. Let $\mathcal{X}$ denote the set of terminal states. We assume that all states in $\mathcal{S}$ are reachable from $\mathbf{s}_0$ through a sequence of actions, though not necessarily by a unique sequence. A *complete trajectory* is a sequence of states $\tau = (\mathbf{s}_0 \rightarrow \mathbf{s}_1 \rightarrow \cdots \rightarrow \mathbf{s}_n)$, where $\mathbf{s}_n \in \mathcal{X}$, and each pair of consecutive states is connected by an action, i.e., $\forall i \, (\mathbf{s}_i, \mathbf{s}_{i+1}) \in \mathcal{A}$. A *policy* in this MDP defines a distribution $P_F(\mathbf{s}'|\mathbf{s})$ for each non-terminal state $\mathbf{s} \in \mathcal{S} \setminus \mathcal{X}$, specifying the probability of transitioning to the next state $\mathbf{s}'$ in a single action. The policy induces a distribution over complete trajectories as follows: $P_F(\mathbf{s}_0 \rightarrow \mathbf{s}_1 \rightarrow \cdots \rightarrow \mathbf{s}_n) = \prod_{i=0}^{n-1} P_F(\mathbf{s}_{i+1} \mid \mathbf{s}_i)$. The marginal distribution over terminal states, denoted $P_F^\top$, is the distribution on $\mathcal{X}$ induced by the policy over all complete trajectories. It may be computationally intractable to compute $P_F^\top$ directly, as $P_F^\top(\mathbf{x}) = \sum_{\tau \rightarrow \mathbf{x}} P_F(\tau)$, where the sum is taken over all complete trajectories that terminate at state $\mathbf{x}$.

## 3 Related Work

Unlike Reinforcement Learning (RL), where the goal is typically to maximize the expected reward by learning a deterministic policy (Mnih et al., 2015; Lillicrap et al., 2015; Haarnoja et al., 2017; Fujimoto et al., 2018; Haarnoja et al., 2018), GFlowNets aim to learn a stochastic policy for generating composite objects $x$ with probability proportional to the reward function $R(x)$. This is particularly

useful in real-world tasks where diversity in solutions is crucial, such as recommender systems (Kunaver & Požrl, 2017), drug discovery (Bengio et al., 2021a; Jain et al., 2022a), and sampling causal models from a Bayesian posterior (Deleu et al., 2022). However, existing GFlowNet approaches (Bengio et al., 2021a; Malkin et al., 2022; Madan et al., 2022) have primarily been developed for deterministic environments, where state transitions are fixed. In real-world applications, stochasticity in state transitions is common, presenting significant challenges for GFlowNets. Deterministic GFlowNet methods can fail to model the correct state visitation distribution under stochastic transitions. For instance, in the presence of stochastic dynamics, standard GFlowNets may learn incorrect probabilities for visiting states, which do not align with the ideal distribution. This mismatch occurs because existing methods do not properly account for randomness in state transitions. To address these limitations, we propose a novel approach called **KL Divergence Optimization with Entropy-Ratio Estimation for Stochastic GFlowNets**. Our method introduces a KL divergence objective that optimizes the policy distribution while incorporating an entropy-ratio estimation mechanism that dynamically balances exploration and exploitation. By adjusting the exploration-exploitation trade-off through entropy-ratio estimation, our method enables GFlowNets to capture the correct state visitation distribution, even in stochastic environments.

Our approach is general and can be applied to different GFlowNet learning objectives. It works by minimizing the divergence between forward and backward policies, ensuring flow consistency across stochastic transitions. The entropy-ratio estimation further enhances robustness by favoring high-entropy states in situations where the environment exhibits higher stochasticity. This approach allows for better mode discovery and improves state visitation coverage in stochastic tasks, such as molecule discovery, biological sequence generation, and other structured object generation tasks.

Our contributions of this paper are as follows:

- We propose **KL Divergence Optimization with Entropy-Ratio Estimation for Stochastic GFlowNets**, a novel approach that addresses the limitations of existing GFlowNet methods in stochastic environments.
- We provide a detailed analysis of how our method optimizes the flow consistency and dynamically adjusts exploration in stochastic transitions, making it suitable for a wide range of stochastic tasks.
- We conduct extensive experiments on benchmark tasks, demonstrating that our method significantly outperforms existing baselines, including Stochastic GFlowNets (SGFN), PPO, SAC and MCMC particularly in complex environments like biological sequence generation.

## 4    DETAILED BALANCE IN STOCHASTIC GFLOWNETS

Detailed balance (DB) is a fundamental principle in GFlowNets, ensuring the alignment between forward and backward policies to maintain the desired state distribution. In stochastic GFlowNets, DB must accommodate the randomness inherent in state transitions, which is crucial for accurately representing the distribution over states under varying conditions.

### 4.1    STOCHASTIC ENVIRONMENTS

Stochastic GFlowNets (Pan et al. (2023)) extend the GFlowNet framework to environments where state transitions are stochastic. These models introduce a decomposition of state transitions into two steps: (1) a deterministic agent action and (2) a stochastic environment transition. This decomposition helps in managing stochastic dynamics but increases the complexity of learning due to the introduction of high variance in training, particularly when combined with trajectory balance objectives. Flow consistency is defined in the forward policy:

$$F(s_t)\pi(a_t|s_t) = \sum_{s_{t+1}} F(s_{t+1})\pi_B((s_t, a_t)|s_{t+1}). \tag{1}$$

This equation highlights the balance of flow at each state by equating the inflow (left-hand side) and outflow (right-hand side). It ensures that the total probability mass flowing out of state $s_t$

via policy $\pi$ matches the backward flow from subsequent states $s_{t+1}$. The consistency is vital for GFlowNets as it stabilizes policy training, ensuring each decision balances the resulting flows in a manner proportional to the overall reward. The stochastic state transitions are then applied to the Detailed-Balance(DB) condition as follow

$$F(s_t)\pi(a_t|s_t)P(s_{t+1}|(s_t, a_t)) = F(s_{t+1})\pi_B((s_t, a_t)|s_{t+1}). \tag{2}$$

This equation explicitly introduces the transition probability $P(s_{t+1}|(s_t, a_t))$, capturing the stochastic nature of moving from state-action pairs $(s_t, a_t)$ to the next state $s_{t+1}$. The need for this formulation arises because stochastic transitions introduce variability that must be accounted for in both forward and backward policies to ensure a robust and proportional sampling distribution. This representation ensures that the detailed balance condition holds, preserving proportional flows between forward and backward states, critical for maintaining the integrity of GFlowNets in stochastic environments.

### 4.2 FROM DETAILED BALANCE TO KL DIVERGENCE WITH ENTROPY-RATIO ESTIMATION

To transform the detailed balance equation into a practical training objective, we express it as a KL divergence minimization problem by incorporating entropy ratio density estimation:

Given:

$$P(s_{t+1}|(s_t, a_t)) = \frac{H_{high}(s_{t+1})}{\gamma H_{high}(s_{t+1}) + (1-\gamma)H_{low}(s_{t+1})}, \tag{3}$$

where $H_{high}(s_{t+1})$ and $H_{low}(s_{t+1})$ represent the densities related to high and low entropy states, respectively and $0 < \gamma < 1$. We can rewrite the detailed balance in terms of this density ratio. GFlowNet detailed balance is an off-policy algorithm that leverages training data from a variety of distributions. Specifically, we can reframe the detailed balance objective from (Eq. 2) given the environemnt dynamic (Eq. 3) into a KL divergence formulation

$$\min_\theta D_{KL}\left(\pi_B((s_t, a_t)|s_{t+1})\left\|\frac{F((s_t, a_t))\frac{H_{high}(s_{t+1})}{\gamma H_{high}(s_{t+1})+(1-\gamma)H_{low}(s_{t+1})}}{F(s_{t+1})}\right.\right). \tag{4}$$

The KL divergence can also be expressed as a summation over state-action pairs for policy $\pi_B$:

$$D_{KL} = \sum_{s,a,s'} \pi_B((s,a)|s')\left(\log \pi_B((s,a)|s') - \log F(s,a) - \log \frac{H_{high}(s')}{\gamma H_{high}(s') + (1-\gamma)H_{low}(s')} + \log F(s')\right). \tag{5}$$

This summation highlights the direct contribution of state-action pairs, incorporating entropy ratio estimations in the policy optimization process.

## 5 DYNAMICS LOSS USING CROSS ENTROPY WITH ENTROPY-RATIO ESTIMATION

### 5.1 DYNAMICS LOSS

The dynamics loss is a crucial component that aligns the model's predictions with the empirical state transitions observed in stochastic environments. By integrating entropy-ratio estimations, this loss function effectively adjusts the weight given to transitions based on their uncertainty, captured through entropy measures. High-entropy transitions correspond to exploratory actions that increase state visitation diversity, while low-entropy transitions focus on consolidating high-reward paths, aiding exploitation.

**Deriving the Dynamics Loss** (Mohammadpour et al. (2024)) defines the flow entropy which is strictly concave function as

$$H(\pi) = E\left[\sum_{t=0}^{T-1} H(\pi(\cdot|s_t))\right] = \sum_{s\in S} \mu_\pi(s)H(\pi(\cdot|s)), \tag{6}$$

where $H(\pi(\cdot|s)) = -\sum_{a\in A(s)} \pi(a|s)\log\pi(a|s)$ measures the randomness of actions at state $s$. The term $\mu_\pi(s)$ represents the state visitation frequency under the policy $\pi$, denoting how often the state $s$ is encountered when following $\pi$. The flow entropy is calculated as a weighted sum of the entropy of the policy at each state, where $\mu_\pi(s)$ acts as the weighting factor. This approach ensures that states visited more frequently have a greater influence on the overall entropy, which is essential for analyzing the exploration behavior of the policy over time. We express the density ratio entropy for a given $\gamma$ as:

$$r_\gamma(s) = \frac{H_{high}(s)}{\gamma H_{high}(s) + (1-\gamma)H_{low}(s)}, \tag{7}$$

The functional form of entropy (see section 9 for more details) at state $s$ is defined $H_{\text{high}}(s) = \exp\left(-\beta_{\text{high}} \cdot H(s)\right)$, $H_{\text{low}}(s) = \exp\left(-\beta_{\text{low}} \cdot H(s)\right)$, where $\beta_{\text{high}}$ and $\beta_{\text{low}}$ are scaling factors that control the influence of entropy on exploration and exploitation. States with higher entropy contribute more to $H_{\text{high}}(s)$, while states with lower entropy contribute more to $H_{\text{low}}(s)$. This allow to adjust the probability of each state-action pair based on the weighted contributions of high and low entropy states. The dynamics loss, incorporating this entropy ratio, is derived as:

$$\mathcal{L}(\gamma) = -\sum_{s\in S, a\in A(s)} \mu_\pi(s)H(\pi(\cdot|s))\left(\log r_\gamma(s) + (1-\gamma)(1 - H(\pi(\cdot|s)))\log(1 - r_\gamma(s))\right). \tag{8}$$

This loss penalizes deviations from expected entropy-weighted transitions, pushing the policy to optimize flows that balance exploration with exploitation.

$\gamma$ **in Exploration vs. Exploitation Trade-off**: The parameter $\gamma$ plays a pivotal role in managing the trade-off between exploration and exploitation by modulating the influence of high and low entropy states in the transition dynamics. High values of $\gamma$ emphasize high-entropy transitions, favoring exploration by allowing the policy to sample diverse actions and visit more states. This promotes the discovery of new modes, avoiding local optima by spreading the probability mass across a wider set of states. Conversely, lower values of $\gamma$ increase the influence of low-entropy transitions, focusing on exploitation by reinforcing actions that lead to predictable and high-reward states. This controlled trade-off ensures that the policy can balance between exploring new opportunities and exploiting known profitable actions, directly impacting state visitation patterns and the robustness of the learned policy. We provide the algorithm of our proposed method in algorithm 1.

---

**Algorithm 1** KL Divergence Optimization with Entropy-Ratio Estimation for Stochastic GFlowNets

---

1: Initialize policy parameters $\theta$, environment dynamics, and $\gamma$ (exploration-exploitation trade-off).
2: **for** each episode **do**
3:     Initialize state $s_0$.
4:     **while** not in terminal state $s_T$ **do**
5:         Sample action $a_t \sim \pi_\theta(a|s_t)$.
6:         Transition to next state $s_{t+1} \sim P(s_{t+1}|s_t, a_t)$.
7:         Compute reward $R(s_{t+1})$.
8:         Compute entropy ratio for state $s_{t+1}$:

$$r_\gamma(s_{t+1}) = \frac{H_{\text{high}}(s_{t+1})}{\gamma H_{\text{high}}(s_{t+1}) + (1 - \gamma) H_{\text{low}}(s_{t+1})}$$

9:     **end while**
10:    Minimize the KL divergence:

$$D_{KL} = \sum_{s,a,s'} \pi_\theta(s, a|s') \left[\log \pi_\theta(s, a|s') - \log F(s, a) - \log r_\gamma(s')\right]$$

11:    Adaptively update $\gamma$:

$$\gamma_{t+1} = \gamma_t + \eta(\text{Var}(R(s_{t+1})) - \text{Var}(R(s_t)))$$

12: **end for**

---

**Practical Approximation of Dynamics Loss** To make this loss computationally tractable, we approximate it by discretizing the state-action pairs:

$$\mathcal{L}(\theta, \gamma) = -\frac{1}{N} \sum_{n=1}^{N} \left(z_n H(\pi_\theta(a_n|s_n)) \log r_\gamma(s_n) + (1 - z_n) H(\pi_\theta(a_n|s_n)) \log(1 - r_\gamma(s_n))\right),$$

(9)

where $z_n$ denotes a binary classification that captures the occurrence of state-action pairs. This practical form illustrates how high entropy (exploration) encourages broader state sampling, while low entropy (exploitation) consolidates high-value trajectories, effectively guiding the policy.

**Effect of Entropy on Loss Dynamics** High entropy in state-action pairs incentivizes the policy to discover new modes by exploring diverse states, preventing overfitting to high-reward areas and maintaining broad state coverage. Low entropy, conversely, focuses on refining the policy toward known high-reward paths, ensuring stability in high-reward regions. The balance between these influences is controlled by $\gamma$, enabling adaptive policy adjustments that enhance overall robustness.

## 6   ANALYSIS OF $r_\gamma(s)$, STOCHASTICITY LEVEL $\alpha$, AND DYNAMIC ADJUSTMENT OF $\gamma$

The entropy ratio $r_\gamma(s)$ in Algorithm 1 is defined as:

$$r_\gamma(s) = \frac{H_{\text{high}}(s)}{\gamma H_{\text{high}}(s) + (1 - \gamma) H_{\text{low}}(s)}$$

where $H_{\text{high}}(s) = \exp(-\beta_{\text{high}} \cdot H(s))$ and $H_{\text{low}}(s) = \exp(-\beta_{\text{low}} \cdot H(s))$. This ratio helps balance the influence of high-entropy and low-entropy states, effectively managing the exploration-exploitation trade-off. The parameter $\gamma$ plays a critical role in determining this balance.

The stochasticity level is controlled by the parameter $\alpha$, which represents the probability of taking random actions. A higher value of $\alpha$ indicates more random (exploratory) behavior, while a lower $\alpha$ implies more deterministic (exploitative) behavior. When $\alpha$ is high, the system favors exploratory

actions, which means that the entropy $H(s)$ of state $s$ tends to be higher, affecting $H_{\text{high}}(s)$ and $H_{\text{low}}(s)$ accordingly. Since $H(s)$ is high, $H_{\text{high}}(s)$ will have a moderate to low value due to the negative exponential term, whereas $H_{\text{low}}(s)$ will be comparatively smaller. Consequently, $r_\gamma(s)$ will be influenced more by the high-entropy component, encouraging diverse state visitation. Conversely, when $\alpha$ is low, the policy is more deterministic, leading to lower entropy in state visitation. In this case, $H(s)$ is low, which increases $H_{\text{high}}(s)$ and $H_{\text{low}}(s)$, but the low-entropy component $H_{\text{low}}(s)$ has a greater influence. The ratio $r_\gamma(s)$ will thus emphasize exploitation, focusing more on high-reward, low-entropy states. The parameter $\gamma$ is updated dynamically in Algorithm 1 to balance exploration and exploitation, depending on the observed variance in reward distributions. The adaptive update rule is given by:

$$\gamma_{t+1} = \gamma_t + \eta(\text{Var}(R(s_{t+1})) - \text{Var}(R(s_t)))$$

where $\eta$ is the learning rate controlling the adjustment step, and $\text{Var}(R(s_t))$ and $\text{Var}(R(s_{t+1}))$ are the reward variances observed before and after transitioning from state $s_t$ to state $s_{t+1}$. The dynamic adjustment of $\gamma$ is directly influenced by the change in reward variance across state transitions, which reflects the level of uncertainty or stochasticity present in the environment.

When the variance in rewards is high between two successive states ($\text{Var}(R(s_{t+1})) > \text{Var}(R(s_t))$), this implies a high level of uncertainty in the environment. In this case, the update rule for $\gamma$ will increase $\gamma$. This increases the weight on $H_{\text{high}}(s)$ in $r_\gamma(s)$, effectively promoting exploration to gather more information about the uncertain environment. As $\gamma$ increases, $r_\gamma(s)$ is driven more by the high-entropy component, resulting in more diverse state visitation and potentially discovering novel, high-reward paths.

If the reward variance is low ($\text{Var}(R(s_{t+1})) < \text{Var}(R(s_t))$), this implies that the environment is more predictable, and the agent is starting to identify stable, rewarding actions. Consequently, the update rule will decrease $\gamma$, which reduces the weight on $H_{\text{high}}(s)$ and increases the influence of $H_{\text{low}}(s)$, making $r_\gamma(s)$ emphasize low-entropy states. This results in more exploitative behavior, allowing the agent to take advantage of known high-reward actions and converge to a stable, deterministic policy.

The combined effect of $\alpha$ and $\gamma$ on $r_\gamma(s)$ can be understood as follows. When both $\alpha$ and $\gamma$ are high, the agent strongly emphasizes exploration. A high $\alpha$ leads to more random actions, while a high $\gamma$ results in $r_\gamma(s)$ favoring high-entropy states. Together, this results in a broad exploration of the state space, potentially identifying diverse high-reward regions. Conversely, when $\alpha$ and $\gamma$ are both low, the agent's behavior becomes more deterministic and exploitative. The low value of $\alpha$ leads to fewer random actions, while a lower $\gamma$ results in $r_\gamma(s)$ putting greater emphasis on low-entropy states, leading to stable exploitation of known rewards.

The dynamic adjustment of $\gamma$ enables the model to be adaptive based on the stochasticity of the environment. During early stages of training, when the environment is largely unexplored and reward variance is high, $\gamma$ increases, promoting exploration. As training progresses, if reward variance decreases, suggesting the agent has identified promising regions of the state space, $\gamma$ decreases, and the model shifts towards exploitation. This approach ensures a balanced and adaptive exploration-exploitation strategy tailored to the dynamics of the environment. By using the dynamic variance-based adjustment, the agent can systematically adapt $\gamma$ to improve learning efficiency and reward maximization.

## 7 EXPERIMENTS

In this section, we conduct extensive experiments to investigate the following key questions: i) How much can KL Divergence Optimization with Entropy-Ratio Estimation improve the performance of Stochastic GFlowNets over standard GFlowNets in the presence of stochastic transition dynamics? ii) Can our method scale to more complex and challenging tasks, such as generating biological sequences and what is the effect of stochasticity level on its performance?

## 7.1 GRIDWORLD

### 7.1.1 EXPERIMENTAL SETUP

We begin by conducting a series of experiments in the GridWorld task, originally introduced in Bengio et al. (2021a), to evaluate the effectiveness of GFlowNets optimization scheme. An illustration of the task, with a grid size of $H \times H$, is shown in Figure 1. At each time step, the agent selects an action to navigate the grid. The available actions include increasing a coordinate, and a stop operation, which terminates the episode and ensures the underlying Markov decision process (MDP) forms a directed acyclic graph (DAG). The agent receives a reward $R(x)$, as defined in Bengio et al. (2021a), when a trajectory reaches a terminal state $x$. The reward function $R(x)$ has four distinct modes, located in the corners of the map (Figure 1). The agent's ob-

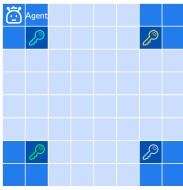

Figure 1: The GridWorld environment. The agent starts at the top-left corner and receives the highest reward at the four dark blue positions near the corners (with keys), lower rewards at the $2 \times 2$ squares near the corners, and even lower rewards at the lighter blue positions. Different grid sizes $H$ and noise levels $\alpha$ can be explored.

jective is to model the target reward distribution and capture all reward modes. The shade of color reflects the magnitude of the rewards, with darker colors indicating higher rewards. We introduce stochasticity into the environment by adopting the transition dynamics from Machado et al. (2017) and Yang et al. (2022). Specifically, with probability $1 - \alpha$, the environment follows the selected action, but with probability $\alpha$, a uniformly chosen random action is executed (leading to slips or missteps to neighboring regions, as shown in Figure 1).

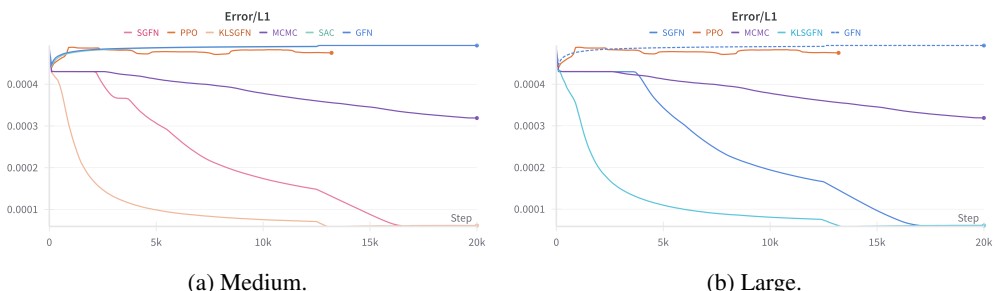

(a) Medium.

(b) Large.

Figure 2: Comparison results of $L_1$ error in GridWorld for varying map sizes.

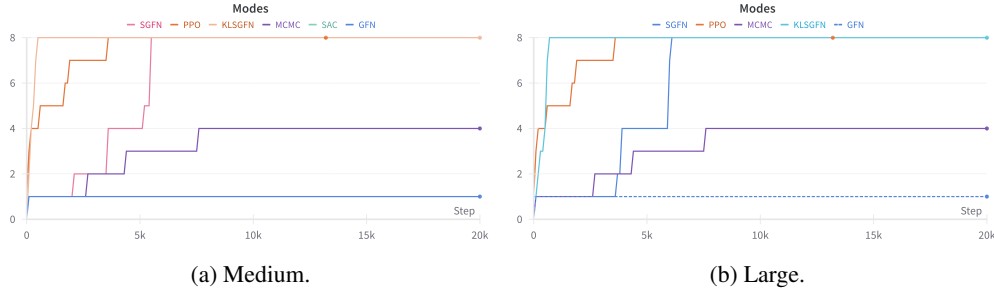

(a) Medium.

(b) Large.

Figure 3: Comparison results of the number of modes captured during training in GridWorld with varying map sizes.

We compare the performance of KL Divergence Optimization with Entropy-Ratio Estimation against vanilla GFlowNets, which are trained using trajectory balance (TB) Malkin et al. (2022), Stochastic GFlowNets (SGFN) Pan et al. (2023), as well as other methods like Metropolis-Hastings

MCMC Xie et al. (2021) and PPO Schulman et al. (2017). The evaluation is based on the empirical $L_1$ error, which measures the difference between the true reward distribution $p(x) = \frac{R(x)}{Z}$ and the estimated distribution $\pi(x)$, derived by repeated sampling and frequency counting of visits to all possible states $x$. Additionally, we compare the methods by counting the number of modes captured during training. Each algorithm is run with five different random seeds, and we report the mean of performance. The implementation details for each baseline are based on open-source code[1].

### 7.1.2 Performance Comparison

We now evaluate the effectiveness of KL Divergence Optimization with Entropy-Ratio Estimation across different map sizes and stochasticity levels in the GridWorld environment.

**Varying map sizes.** Figure 2 shows the empirical $L_1$ error for each method in GridWorld (with stochasticity level $\alpha = 0.25$) as the grid size increases. The results demonstrate that MCMC struggles with larger grids, and GFN fails to converge. Additionally, the performance of TB degrades significantly as the grid size grows, likely due to higher gradient variance, as suggested by Madan et al. (2022). In contrast, our proposed method (KL Divergence Optimization with Entropy-Ratio Estimation) consistently achieves the lowest $L_1$ error and converges faster than all baselines, including stochastic GFlowNets with DB-objective and Vanilla GFlowNets with TB-objective.

### 7.2 Biological Sequence Generation

In biological sequence generation, the objective is to discover sequences with optimal properties by maximizing a reward function corresponding to specific biological traits. This task is particularly challenging due to the inherent complexity and stochasticity present in biological environments. For example, in the task of generating DNA sequences, the objective might be to find sequences that exhibit high binding affinity to a particular transcription factor.

To demonstrate the efficacy of our proposed method, we evaluate it on the TFBind8 task, where the goal is to generate strings of nucleotides (e.g., DNA sequences of length 8). Conventionally, such tasks are modeled using an autoregressive Markov Decision Process (MDP). However, we utilize a prepend-append MDP (PA-MDP), where actions involve adding tokens (e.g., nucleotides) either to the beginning or the end of a partial sequence. The reward function, in this case, measures the DNA binding affinity to a human transcription factor, providing feedback on the sequence's fitness.

The complexity of the biological sequence generation problem lies in handling the vast combinatorial search space and the stochastic nature of sequence interactions with biological targets. By introducing stochasticity into the environment through transition dynamics, our method dynamically balances exploration and exploitation using KL divergence optimization and entropy-ratio estimation, which improves the robustness of the generated sequences and ensures better mode discovery.

**TFBind8.** Our goal is to generate a string of length 8 of nucleotides. Though an autoregressive MDP is conventionally used for strings, we use a prepend-append MDP (PA-MDP) Shen et al. (2023), in which the action involves either adding one token to the beginning or the end of a partial sequence. The reward is a DNA binding affinity to a human transcription factor Trabucco et al. (2022).

### 7.2.1 Quality of Rewards

The quality of the rewards during the sequence generation process is directly influenced by the adaptive tuning of the parameter $\gamma$, which balances exploration and exploitation through $r_\gamma(s)$. As shown in Figure 4, the quality of rewards for varying stochasticity levels ($\alpha = 0.25, 0.50, 0.75$) demonstrates that our approach maintains high reward quality, even under increased stochasticity. Specifically, as $\alpha$ increases, representing a more stochastic environment, existing methods such as Stochastic GFlowNets (SGFN) and standard GFlowNets struggle to maintain reward quality. In contrast, our method, which incorporates entropy-ratio estimation, successfully adapts to the increased randomness by adjusting $\gamma$ dynamically. The entropy ratio $r_\gamma(s)$ favors exploratory actions in uncertain environments, thereby allowing the agent to discover new high-reward sequences while maintaining robustness.

---

[1] https://github.com/GFNOrg/gflownet

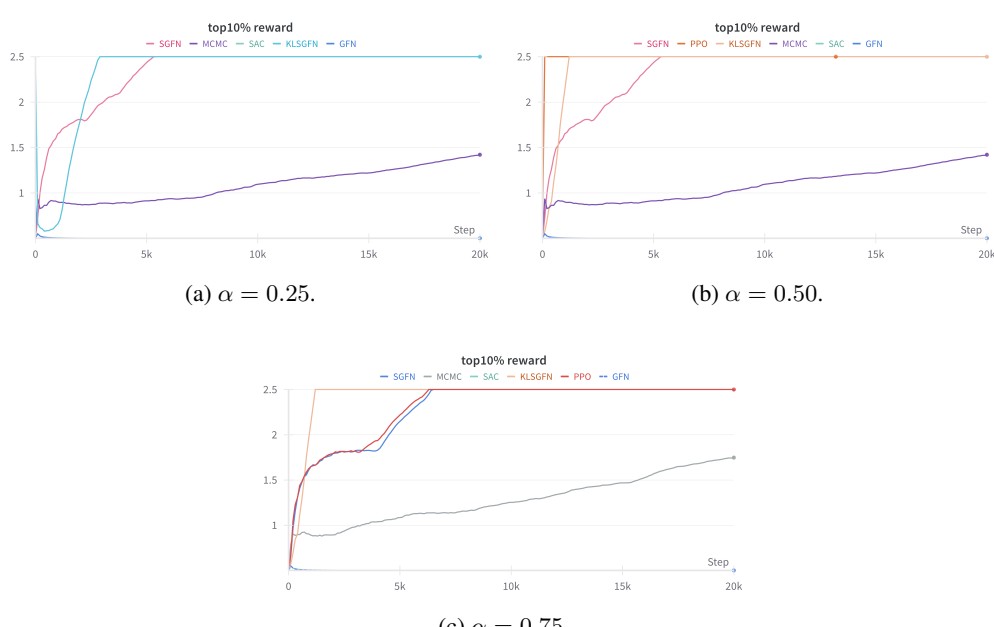

Figure 4: Comparison results of the quality of rewards captured during training for TFBIND experiment for different levels of stochasticity.

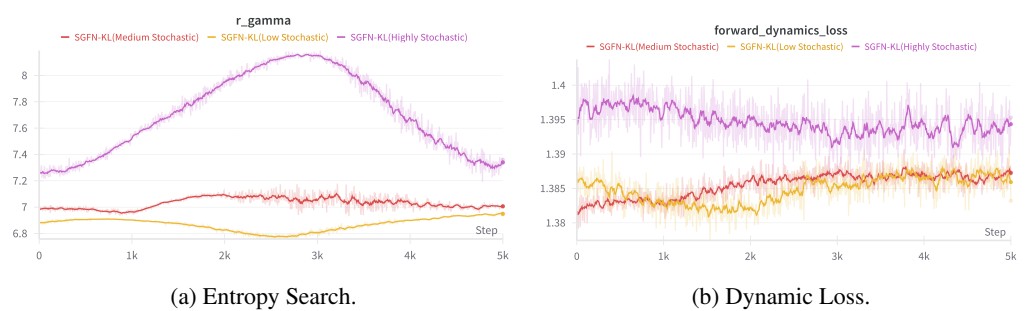

Figure 5: Comparison results of the Entropy Search and Dynamic loss behavior captured during training for TFBIND experiment for different levels of stochasticity.

### 7.2.2 EFFECT OF $r_\gamma(s)$ ON ENTROPY SEARCH AND MODE DISCOVERY

The entropy ratio $r_\gamma(s)$ plays a critical role in controlling the balance between high-entropy (exploratory) and low-entropy (exploitative) states. Figure 5(a) shows impact of dynamically adjusting $\gamma$ values on entropy-based search and mode discovery. During the initial phases of training, when $\alpha$ is high, $\gamma$ also increases, leading to a greater emphasis on high-entropy components ($H_{high}(s)$). This emphasis encourages exploration by allowing the agent to sample a broader set of actions and visit more states, ultimately leading to better mode discovery. This behavior is evident in Figure 5(b), where higher values of $\gamma$ result in the agent discovering more unique modes, thus avoiding premature convergence to suboptimal sequences. As training progresses and reward variance decreases, $\gamma$ is adaptively reduced, shifting the balance towards low-entropy states ($H_{low}(s)$), which promotes the exploitation of known high-reward sequences.

### 7.2.3 DYNAMIC LOSS AND ADAPTATION

Figure 5(b) illustrates the behavior of dynamic loss during the training process for different stochasticity levels. The dynamics loss, incorporating entropy-ratio estimations, penalizes deviations from the expected entropy-weighted transitions, effectively aligning the model's predictions with observed state transitions. The dynamic update of $\gamma$ follows the rule:

$$\gamma_{t+1} = \gamma_t + \eta \left( \text{Var}(R(s_{t+1})) - \text{Var}(R(s_t)) \right)$$

where $\eta$ is the learning rate. When the variance in rewards between two consecutive states ($\text{Var}(R(s_{t+1}))$ and $\text{Var}(R(s_t))$) is high, indicating a high level of environmental uncertainty, $\gamma$ is increased to favor exploration. This adjustment causes $r_\gamma(s)$ to prioritize high-entropy states, thereby enhancing exploration capabilities. Conversely, as the environment becomes more predictable and reward variance decreases, $\gamma$ is reduced, leading $r_\gamma(s)$ to emphasize low-entropy states and focus on exploitation. This dynamic adaptation helps in maintaining a balance between discovering new high-reward sequences and exploiting known sequences, ultimately ensuring robustness and efficiency in learning.

### 7.2.4 ENTROPY SEARCH ON DYNAMIC STOCHASTICITY

Figure 5 depicts the interaction between $\gamma$ and the stochasticity level $\alpha$ during entropy-based search. When $\gamma$ is high, the model emphasizes exploring less-visited regions of the sequence space, which is especially effective under lower stochasticity ($\alpha = 0.25$), where the environment is relatively predictable. As $\alpha$ increases, indicating more randomness in the environment, the exploration becomes more erratic. Our method, by dynamically adjusting $\gamma$, manages to stabilize the exploration process even under higher stochasticity, ensuring a balanced trade-off between discovering novel high-reward sequences and exploiting already discovered optimal sequences.

### 7.2.5 IMPACT ON PERFORMANCE AND SEARCH EFFICIENCY

The dynamic interaction between $\gamma$ and $\alpha$ plays a pivotal role in enhancing search efficiency in biological sequence generation. The parameter $\gamma$, by modulating the entropy ratio $r_\gamma(s)$, allows the model to adjust its behavior based on the stochastic nature of the task, encouraging exploration when necessary and promoting reliable exploitation once high-quality sequences are found. On the other hand, $\alpha$ controls the environment's stochasticity, influencing how the model handles uncertain transitions. Together, these parameters ensure an optimal balance between exploration and exploitation, leading to improved performance in generating biological sequences with desirable properties, as seen in Figure 4 and Figure 5.

## 8 CONCLUSION

In this paper, we introduced a novel methodology, *KL Divergence Optimization with Entropy-Ratio Estimation for Stochastic GFlowNets*, which effectively extends GFlowNets to more complex and realistic stochastic environments, where existing GFlowNet approaches tend to underperform. Our method not only learns the GFlowNet policy but also incorporates entropy-ratio estimation to dynamically balance exploration and exploitation, making it more robust to stochastic transitions.

We conducted extensive experiments on standard GFlowNet benchmark tasks augmented with stochastic transition dynamics, demonstrating that our method significantly outperforms previous methods in terms of both mode discovery and state visitation coverage. The results show that by leveraging KL divergence optimization and entropy-ratio estimation, our approach can better handle the stochasticity in environments, leading to more efficient and accurate policy learning.

Future research could explore advanced model-based approaches for approximating transition dynamics in stochastic environments. Additionally, our method opens new possibilities for applying GFlowNets to other challenging real-world tasks, such as biological sequence generation and molecule discovery, where stochasticity plays a key role.

## 9 ACKNOWLEDGEMENTS

The authors would like to express their gratitude to colleagues and collaborators for their valuable insights and feedback on this work. We also appreciate the support from the research community for providing open-source resources and discussions that contributed to the development of our method. Finally, we would like to thank our respective institutions and funding agencies for their continued support and encouragement in advancing this research.

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

## ARCHITECTURAL DETAILS FOR EXPERIMENTS

### GRIDWORLD EXPERIMENTS

To reproduce the GridWorld experiments, we used the following **network architecture**:

- **Policy Network**:
  - **Input Layer**: The state representation is a 2D grid encoded as a flattened vector of size $H \times H$, where $H$ is the grid size.
  - **Hidden Layers**:
    * **Layer 1**: Fully connected layer with 128 neurons, ReLU activation.
    * **Layer 2**: Fully connected layer with 64 neurons, ReLU activation.
  - **Output Layer**: Outputs the probabilities over all possible actions, implemented as a softmax layer to ensure a valid probability distribution.
- **Training Details**:
  - **Optimizer**: Adam optimizer with a learning rate of 0.001.
  - **Batch Size**: 32.
  - **Exploration Parameter** ($\gamma$): Initially set to 0.5 and adjusted adaptively during training based on the observed variance in the state-action values.
  - **Entropy Regularization**: An additional entropy regularization term is added to the loss function to encourage exploration, with a weight of 0.01.

### BIOLOGICAL SEQUENCE GENERATION (TFBIND8)

For the TFBind8 biological sequence generation experiment, we used the following **architecture**:

- **Policy Network**:
  - **Input Layer**: The input is a partial sequence of nucleotides represented as a one-hot encoded vector. For sequences of length 8, the input size is $8 \times 4$ (since each nucleotide can be one of 4 bases).
  - **Embedding Layer**: An embedding layer maps the one-hot encoded representation to a continuous vector space of dimension 16.
  - **LSTM Layer**: A single LSTM layer with 128 hidden units is used to capture dependencies between different positions in the sequence.
  - **Fully Connected Layer**: The LSTM output is passed through a fully connected layer with 64 neurons and ReLU activation.
  - **Output Layer**: Outputs the probability distribution over the four nucleotides for the next position, implemented as a softmax layer.
- **Training Details**:
  - **Optimizer**: Adam optimizer with a learning rate of 0.0005.
  - **Batch Size**: 64.
  - **Sequence Augmentation**: During training, random noise is added to the nucleotide embeddings to simulate stochasticity in biological environments.
  - **Entropy Regularization**: To ensure mode discovery, we add an entropy regularization term with a weight of 0.05.

## DETAIL ON FUNCTIONAL FORM AND PARAMETER TUNING

### FUNCTIONAL FORM OF ENTROPY

The entropy for a given state $s$ is defined as:

$$H(s) = -\sum_a \pi(a|s) \log \pi(a|s)$$

FORM OF $H_{\text{HIGH}}(s)$ AND $H_{\text{LOW}}(s)$

Using the entropy $H(s)$, the quantities $H_{\text{high}}(s)$ and $H_{\text{low}}(s)$ are defined as follows:

$$H_{\text{high}}(s) = \exp\left(-\beta_{\text{high}} \cdot H(s)\right), \quad H_{\text{low}}(s) = \exp\left(-\beta_{\text{low}} \cdot H(s)\right)$$

Where $\beta_{\text{high}}$ and $\beta_{\text{low}}$ are **scaling factors** that adjust sensitivity to entropy. This formulation ensures that states with higher entropy contribute more to $H_{\text{high}}(s)$, while low-entropy states contribute more to $H_{\text{low}}(s)$.

TUNING $\beta_{\text{HIGH}}$ AND $\beta_{\text{LOW}}$

The parameters $\beta_{\text{high}}$ and $\beta_{\text{low}}$ control the influence of **high-entropy** and **low-entropy** states in balancing exploration and exploitation using the following strategy:

- Dynamic Tuning: Initially, $\beta_{\text{high}} = 0.4$ and $\beta_{\text{low}} = 0.6$ are used (as a warm-up). If the model converges too quickly or fails to explore effectively, $\beta_{\text{high}}$ can be increased or $\beta_{\text{low}}$ decreased to encourage more exploration. If reward variance is high and convergence is not achieved, increasing $\beta_{\text{low}}$ can help focus on exploiting high-reward actions.

Monitoring metrics like **reward variance** and **state visitation** allows adaptive tuning of $\beta$ values to achieve an optimal exploration-exploitation balance:

$$H_{\text{high}}(s) = \exp(-\beta_{\text{high}} \cdot H(s)), \quad H_{\text{low}}(s) = \exp(-\beta_{\text{low}} \cdot H(s))$$

ADAPTIVE UPDATE OF $\gamma$

To balance exploration and exploitation, the parameter $\gamma$ is adapted dynamically during training:

$$\gamma_{t+1} = \gamma_t + \eta(\text{Var}(R(s_{t+1})) - \text{Var}(R(s_t)))$$

Where $\eta$ is the **learning rate** that controls the adjustment of $\gamma$ based on reward variance. This dynamic adjustment ensures that $\gamma$ evolves in response to observed changes in environmental stochasticity.

