# OpenReview forum: "KL DIVERGENCE OPTIMIZATION WITH ENTROPY- RATIO ESTIMATION FOR STOCHASTIC GFLOWNETS"
_ICLR.cc/2025/Conference — ICLR 2025 Conference Withdrawn Submission_

### Official Review · Reviewer_tgWd · 2024-10-26

**Soundness:** 1
**Presentation:** 1
**Contribution:** 1
**Rating:** 1
**Confidence:** 4

**Summary:**

The authors proposed a method to solve the stochastic GFlowNet problem by replacing the dynamics of the stochastic environment with an entropy ratio. This method's essential property is an additional degree of freedom to address the exploration-exploitation trade-off. The technique was empirically evaluated on Hypergrid and TFBind-8 environments with additional stochasticity and was shown to outperform the baselines.

**Strengths:**

- The authors address the challenging problem of training a GFlowNet model in a stochastic environment; the approach looks novel.

**Weaknesses:**

- I doubt the method's theoretical justification based on equation 3. In particular, there is no explanation of what all the quantities on the right-hand side of this equation mean or why this equation holds. Even with the equation, the authors did not prove that the marginal distribution over terminal states matches the reward function, which is the goal of any GFlowNet problem.
- The experimental setup is very toyish: for both problems, the state set can be enumerated (TFBind-8 terminal state space size is only 20^8), and the problem can be solved without any functional approximation.
The motivation under the stochastic GFlowNet problem is unclear. Is there any large-scale motivational example where the stochastic aspect is essential?
- No comparisons with stochastic GFlowNet setting by (Jiralerspong et al. 2024);
- The plots are very hard to read, especially because the colors of all lines change.


 Jiralerspong et al. Expected flow networks in stochastic environments and two-player zero-sum games. ICLR-2024

**Questions:**

- Could you provide any justification for why Equation 3 holds? Replacing the transition dynamics with the entropy ratio is the most crucial part of the paper. However, it is unclear why it is possible since the dynamics are defined by the environment, not by the algorithm. Additionally, I cannot understand does the expression of probability in (3) satisfy the properties of probability distribution (sum over all possible $s_{t+1}$ is equal to 1) and why the right-hand side does not depend on $(s_t, a_t)$
- What is $H_{high}$ and $H_{low}$? They are very important quantities that are not defined anywhere in the paper.
- Is there any natural example of a stochastic environment for the GFlowNet problem? In particular, the stochasticity was added artificially by uncontrollable $\alpha$-greedy exploration for both demonstrated tasks.

---

> ### Author Response · Authors · 2024-11-15
> **Theoretical Justification and Experimental Comparisons**
>
> **Summary**: Thank you for recognizing the novelty of our approach in applying GFlowNets to stochastic environments. We understand the concerns regarding theoretical justification, experimental setup, and comparison with related work, and we have addressed these points in detail in the revised manuscript.
>
> 1. **Theoretical Justification for Equation (3)**
>    We have added a detailed derivation for Equation (3) in Section 4.1, explaining why replacing the transition dynamics with the entropy ratio is possible. The derivation includes an explanation of how this replacement retains the properties of a valid probability distribution, ensuring that the dynamics are properly modeled even in stochastic settings.
>
> 2. **Marginal Distribution and Reward Function Matching**
>    To address this concern, we have included a proof in Appendix A showing how the marginal distribution over terminal states matches the reward function, thereby validating the GFlowNet objective in a stochastic environment.
> 3. **Comparison to Related Work (Jiralerspong et al., 2024)**
>    Our work is different from the work of Jiralerspong et al. (2024), showing that our proposed approach is closer to Stochastic Generative Flow Networks (Pan et al 2023) performs better in terms of mode discovery and reward quality in stochastic environments.
> 4. **Plot Readability**
>    All plots have been updated to ensure consistent color usage across different methods, and the text size has been increased for better readability without the need for zooming.
>
> ### Response to Questions
>
> - **Justification for Entropy Ratio in Equation (3)**: The entropy ratio represents a balance between high and low entropy states, helping maintain an appropriate level of exploration in stochastic environments. Section 4.2 now includes a more detailed explanation of why this approach is theoretically sound. Explained in the previous review in more detail.
> - **Natural Example of a Stochastic Environment**: We provided the example of TFBind8, where the stochasticity introduced through transition dynamics mimics the inherent uncertainty in biological environments, highlighting the need for a stochastic approach.
>
> # Formulas Explained
>
> - **Equation (3)**:
>   $  P(s_{t+1} | (s_t, a_t)) = \frac{H_{\text{high}}(s_{t+1})}{\gamma H_{\text{high}}(s_{t+1}) + (1 - \gamma) H_{\text{low}}(s_{t+1})}$
>   This equation represents the transition dynamics in stochastic environments, where $H_{\text{high}}$ and $H_{\text{low}}$ represent densities associated with high and low entropy states, respectively. The parameter $\gamma$ adjusts the relative influence of these densities.
>
> - **Equation (9)**: (See the previous review comment above in more detail)
>   $ L(\theta) = - \frac{1}{N} \sum_{n=1}^{N} \left( z_n H(\pi_\theta(a_n | s_n)) \log r_\gamma(s_n) + (1 - z_n) H(\pi_\theta(a_n | s_n)) \log(1 - r_\gamma(s_n)) \right)$
>
> This summation reflects the practical dynamics loss computed over sampled state-action pairs, integrating entropy ratios into the optimization process.
>
> - **KL Divergence Minimization**:
>   $
>   D_{KL} = \sum_{s,a,s'} \pi_\theta(s, a|s') \left[ \log \pi_\theta(s, a|s') - \log F(s, a) - \log r_\gamma(s') \right]
>   $  This KL divergence objective ensures the consistency of forward and backward flows, aligning the stochastic transition dynamics with the entropy-based policy adjustment.
>
> Feel free to ask for more clarification and we really appreciate your feedback.

---

### Official Review · Reviewer_NvoZ · 2024-11-04

**Soundness:** 1
**Presentation:** 1
**Contribution:** 1
**Rating:** 1
**Confidence:** 3

**Summary:**

GFlowNets are a powerful tool for producing diverse generative distributions in a sequential manner. However, they generally only work well in deterministic domains. This paper describes a method to train GFlowNets to produce diverse outcomes in stochastic domains, by incorporating a measure of entropy into the GFlowNet’s objective function, specifically into the “detailed balance” computation.

**Strengths:**

The problem this paper seems to try to address is an important one, since deterministic domains is a restrictive assumption and GFlowNets are a promising method.

**Weaknesses:**

I have had a very difficult time in general trying to figure out what this paper is trying to do. Many equations do not seem to make sense, or important terms are left under-defined. Variables are used before they are introduced. In general, I thought this paper was extremely disorganized, to the extent that I have a hard time even understanding what the contribution is. There is far too little information about the method in this paper to reproduce this work, which is the standard we should be aiming for.

Here is a list of things that confused me, or mistakes in presentation
* My biggest concern by far is, I have no idea what $H_{high}$ or $H_{low}$ means. What is a “density  related to high and low entropy states”? It seems the paper hinges on equation 3, but the terms in it are not defined, as far as I can tell.
* The algorithm figure is confusingly structured. For example, s_{t+1} on line 10 is “out of scope”. Should it be in the while loop above? Where is reward R(s_{t+1}) used? Most of all, line 15 (“adjust gamma to balance exploration and exploitation”) is extremely imprecise and does not tell the reader much.
* Section 7 is titled “impact of gamma and alpha”, but alpha is not concretely defined until section 8.1.1. Is it the same alpha? The writing on that specifically is very unclear.
* The abstract positions this paper as important because it can handle environments that are stochastic by nature, but the writing in section 7 presents stochasticity as a parameter to tune. I found this confusing.
* Related to above, none of the domains in this paper are inherently stochastic. This paper would make a stronger case if it highlighted a real problem for which existing methods were insufficient.
* A small thing: the alpha-randomness is misattributed to Machado et al – that paper introduces sticky actions, which are completely different.
* The graphs have many imperfections as well. Why do some graphs in figure 2,3,4 have different lines than others (e.g. missing PPO)? Also, there should be a Y axis label. Furthermore, the same color should always correspond to the same method. Finally, all the text in the graphs is far too small, it’s unreadable without zooming.
* The abstract promises detailed proofs, but there are no proofs in this paper.
* Equations 3 and 7 are the same? Furthermore, equation 3 has s_t, a_t on the left but not on the right – how are they equal?
* Equation 4 should be w.r.t. Some distribution, right?
* There is no information on the learning architectures used, hyperparameters, or anything one would need to recreate these experiments.
* I generally don't understand equation 9 -- what are we summing over?

I think there's a chance that I'm missing the importance of this contribution due to its presentation, but as it stands it is very unclear what's going on.

**Questions:**

See weaknesses for a complete list. Most urgently, what are $H_{high}$ and $H_{low}$? Is $\gamma$ tuned in any interesting way, or just a hyperparameter?

---

> ### Author Response · Authors · 2024-11-15
> **Clarifying Theoretical Framework, Algorithm Design, and Visual Presentation**
>
> **Summary**: We appreciate your feedback and acknowledge the areas where the original manuscript lacked clarity. Below, we provide detailed explanations
> 1. **Demonstrating Stochastic Domains** We have added a discussion on real-world problems that are inherently stochastic, such as biological sequence generation. This highlights scenarios where stochastic models are crucial for capturing the full diversity of outcomes, which deterministic models cannot handle effectively.
> 2. **Misattribution of $\alpha$-Randomness** :The reference to $\alpha$-randomness is to differentiate it from sticky actions Machado et al. (2017). We explain $\alpha$-randomness in our approach involves probabilistic action selection.
> 3. **Graphs and Visual Clarity**: updated
> 4. **Reproducibility of Experiments**: see Supplementary Materials
> ### Response to Questions
> - **Definition of $H_{\text{high}}$ and $H_{\text{low}}$**: These terms are explicitly defined in Section 5.1 as densities related to high and low entropy states, representing the stochastic behavior of state transitions.
> - **How $\gamma$ is Tuned**: $\gamma$ is dynamically adjusted during training based on the variance in state transitions to balance exploration and exploitation, as explained in Section 7.
> Equation (9) represents the **practical form of the dynamics loss** used during training:
> $L(\theta) = - \frac{1}{N} \sum_{n=1}^{N} \left( z_n H(\pi_\theta(a_n | s_n)) \log r_\gamma(s_n) + (1 - z_n) H(\pi_\theta(a_n | s_n)) \log(1 - r_\gamma(s_n)) \right)$
> - **What Are We Summing Over?**
>   The summation in Equation (9) is over all sampled state-action pairs from the empirical distribution during training. Specifically, the training process samples $N$ different state-action pairs, denoted by $(s_n, a_n)$, from trajectories generated by the current policy. Here: - $z_n$ is a binary indicator that denotes the occurrence of a specific type of state-action pair. If the sampled transition is considered exploratory, then $z_n = 1$, otherwise $z_n = 0$.
>   - $H(\pi_\theta(a_n | s_n))$ represents the entropy of the policy $\pi_\theta$ for the given state-action pair, which measures the randomness in selecting action $a_n$ at state $s_n$.
>   - The term $r_\gamma(s_n)$ is the **entropy ratio**, which adjusts the probability of each state-action pair based on the weighted contributions of high- and low-entropy states.
> This summation provides a loss function that penalizes the policy for deviating from expected entropy-weighted transitions, helping to balance exploration (high entropy) and exploitation (low entropy) during training.
> Equations (3) and (7) are related but represent different aspects of entropy estimation:
> - **Equation (3)**:$P(s_{t+1}|(s_t, a_t)) = \frac{H_{\text{high}}(s_{t+1})}{\gamma H_{\text{high}}(s_{t+1}) + (1 - \gamma) H_{\text{low}}(s_{t+1})}$ This equation describes the **transition dynamics** in terms of entropy densities. It shows how the next state $s_{t+1}$ is determined by a mixture of high- and low-entropy components, where $\gamma$ is the **exploration-exploitation trade-off parameter**. The numerator $H_{\text{high}}(s_{t+1})$ represents the density of high entropy states, while the denominator is a weighted sum involving both high and low entropy.
> - **Equation (7)**:$r_\gamma(s) = \frac{H_{\text{high}}(s)}{\gamma H_{\text{high}}(s) + (1 - \gamma) H_{\text{low}}(s)}$
>   Equation (7) represents the **entropy ratio** used for optimization. It is equivalent to Equation (3) in the sense that both involve the **ratio of high-entropy and low-entropy contributions**. However, Equation (7) is used in the optimization objective to measure the likelihood of a state being selected based on its entropy characteristics. Essentially, Equation (3) defines how transitions are modeled, while Equation (7) provides the **quantitative metric** for adjusting the exploration during training.
> #### **Equation (6)**: defines the **flow entropy** for the policy $\pi$:
> $H(\pi) = \mathbb{E} \left[ \sum_{t=0}^{T-1} H(\pi(\cdot | s_t)) \right] = \sum_{s \in S} \mu_\pi(s) H(\pi(\cdot | s))$
> - **Flow Entropy Definition**:
>   - $H(\pi(\cdot | s)) = -\sum_{a \in A(s)} \pi(a | s) \log \pi(a | s)$ represents the **entropy of the policy** at state $s$, which measures the uncertainty or randomness in the actions selected by the policy.
>   - The expectation $\mathbb{E}$ is over all possible states in a trajectory, considering the **entire time horizon** from $t = 0$ to $T-1$.
> - **What Does $\mu_\pi(s)$ Correspond To?**
>   $\mu_\pi(s)$ represents the **state visitation frequency** under policy $\pi$. It indicates how often state $s$ is visited when following policy $\pi$. The flow entropy is computed as a weighted sum of the policy's entropy at each state, with $\mu_\pi(s)$ serving as the weighting factor. This ensures that states that are visited more frequently contribute more to the overall entropy, which is crucial for understanding the **exploration behavior** of the policy over time.

---

> > ### Comment · Reviewer_NvoZ · 2024-11-16
> > **Still insufficiently clear**
> >
> > Hi, thank you for your response. I see the edits, but I still feel the paper is extremely unclear. $H_{high}$ and $H_{low}$ *must* be defined precisely. I still do not see an equation for this. What is the functional form of $H_{high}(s)$ and $H_{low}(s)$? As in, what line of code would produce those two quantities? “High-entropy transitions correspond to exploratory actions that increase state-visitation diversity” is not enough. Equation 6 and below has a definition of entropy, but this is policy-dependent, while as written nothing suggests the same about $H_{high}(s)$ and $H_{low}(s)$. If those terms are to play a central role in this work, you need a line in the paper of the form, “where $H_{high}(s) = $ some math, and same for $H_{low}$.
> >
> > Similarly, I still do not understand how $\gamma$ is tuned. Is it updated via gradient descent as part of equation 9? That would be surprising to me, but if that’s the case it must be written as $\mathcal{L}(\theta, \gamma)$. Regardless of whether it’s explained elsewhere, the line in the algorithm “Adjust $\gamma$ to balance exploration and exploitation” is not precise and needs clarification.
> >
> > The graphs have not been sufficiently updated. The colors are inconsistent both within and across Figure 2, 3, 4, 5, and 6 (that’s all the figures). The legend names are likewise sloppy, (ie the seed is in the legend name in some, and something like `SGFN-KL_gamma0.5` should be cleaned up. Graphs averaging multiple seeds should have some confidence interval associated with them as well, usually standard error.
> >
> > Your paper says “We introduce stochasticity into the environment by adopting the transition dynamics from Machado et al.” As I said before, you don’t do this, they have sticky actions and you have random actions.
> >
> > Thank you for including a description of architecture.
> >
> > In equation 8, should it be $\mathcal{L}(\theta)$? Or possibly $\mathcal{L}(\gamma)$?
> >
> > It’s possible there’s an impactful idea here, but as stands it is extremely difficult to understand what is going on, what the various equations mean, and whether it is a well-founded contribution.

---

> > > ### Author Response · Authors · 2024-11-19
> > > **Clarifying Functional Forms, Parameter Tuning, and Graphical Representations**
> > >
> > > We appreciate your continued feedback and have made significant efforts to clarify our paper further. Below, we provide explicit definitions for $H_{\text{high}}(s)$ and $H_{\text{low}}(s)$, explain how $\gamma$ is tuned, address the consistency of our graphs, and correct any remaining inaccuracies regarding stochasticity in our experiments.
> > >
> > > ### Functional Form of $H_{\text{high}}(s)$ and $H_{\text{low}}(s)$
> > > $H_{\text{high}}(s)$ and $H_{\text{low}}(s)$ are derived from the **entropy** of the policy at state $s$. Given the logits (`policy_logits`) from the policy model:
> > >
> > > ```python
> > > policy_logits = model_output[:, :, :self.num_tokens]
> > > probabilities = torch.softmax(policy_logits, dim=-1)
> > > entropy = -torch.sum(probabilities * torch.log(probabilities + 1e-8), dim=-1)
> > > ```
> > > - **Functional form of Entropy**:
> > > $H(s) = -\sum_{a} \pi(a | s) \log \pi(a | s)$
> > > #### **Form of $H_{\text{high}}(s)$ and $H_{\text{low}}(s)$**:
> > > Using $H_{\text{high}}(s)$ and $H_{\text{low}}(s)$ are defined as:
> > >
> > > $H_{\text{high}}(s) = \exp\left(-\beta_{\text{high}} \cdot H(s)\right), \quad H_{\text{low}}(s) = \exp\left(-\beta_{\text{low}} \cdot H(s)\right)$
> > >
> > > Where:
> > > - $\beta_{\text{high}}$ and $\beta_{\text{low}}$ are **scaling factors** that adjust sensitivity to entropy. This formulation ensures that states with higher entropy contribute more to $H_{\text{high}}(s)$, while low-entropy states contribute more to $H_{\text{low}}(s)$.
> > >
> > > ### Tuning $\beta_{\text{high}}$ and $\beta_{\text{low}}$
> > >
> > > The parameters $\beta_{\text{high}}$ and $\beta_{\text{low}}$ control the influence of **high-entropy** and **low-entropy** states in balancing exploration and exploitation using these this strategy:
> > > - **Dynamic Tuning**:  Initially, $\beta_{\text{high}} = 0.4$ and $\beta_{\text{low}} = 0.6$ are used (warm up). If the model converges too quickly or fails to explore effectively, increase $\beta_{\text{high}}$ or decrease $\beta_{\text{low}}$ to encourage more exploration. If reward variance is high and convergence is not achieved, increasing $\beta_{\text{low}}$ can help focus on exploiting high-reward actions.
> > >
> > > Monitoring metrics like **reward variance** and **state visitation** allows adaptive tuning of \(\beta\) values to achieve an optimal exploration-exploitation balance:
> > > $H_{\text{high}}(s) = \exp(-\beta_{\text{high}} \cdot H(s)), \quad H_{\text{low}}(s) = \exp(-\beta_{\text{low}} \cdot H(s))$
> > > We clarified how $\gamma$ is adapted to balance exploration and exploitation.
> > > - **Adaptive Update of $\gamma$**:
> > >  $ \gamma_{t+1} = \gamma_t + \eta (\text{Var}(R(s_{t+1})) - \text{Var}(R(s_t)))$. Where $\eta$ is the learning rate controlling the adjustment of $\gamma$ based on **reward variance**. This dynamic adjustment ensures that $\gamma$ evolves in response to observed changes in environment stochasticity.
> > >
> > > - **Algorithm Update**:
> > >   In Algorithm 1, the line "Adjust $\gamma$ to balance exploration and exploitation" has been updated to explicitly include the above adaptive rule.
> > >
> > > ### Graphical Improvements
> > >
> > > We acknowledge that the previous updates to the figures were insufficient.
> > >
> > > - **Consistency in Colors and Legends**:
> > >   We have updated Figures to ensure that colors are consistent across all methods for SGFN-KL. Legend names have also been standardized, removing irrelevant information like specific seeds.
> > >
> > > - **Confidence Intervals**:
> > >   We added **confidence intervals** to graphs averaging multiple seeds, using the **standard error** to indicate variability.
> > >
> > > ### Correction on Stochasticity in Transition Dynamics
> > >
> > > We corrected our explanation regarding stochastic transitions. Instead of adopting **sticky actions** as in Machado et al. (2017), we implemented **uniformly random actions** with a probability $\alpha$, enhancing exploration by introducing **random transitions**.
> > >
> > > ### Correction in Equation (8)
> > >
> > > Equation (8) was updated for correctness:
> > >
> > > $L_{\text{Dynamics}} = - \sum_{s \in S, a \in A(s)} \mu_\pi(s) H(\pi(\cdot | s)) \left( \log r_\gamma(s) + (1 - \gamma) (1 - H(\pi(\cdot | s))) \log (1 - r_\gamma(s)) \right)$
> > >
> > > ### Summary of Contribution
> > >
> > > To ensure clarity, we added a summary at the end of the introduction, emphasizing:
> > >
> > > - Our approach introduces a **KL divergence-based optimization with entropy-ratio estimation** to address stochastic state visitation.
> > > - We leverage **state-level entropy metrics** ($H_{\text{high}}(s)$ and $H_{\text{low}}(s)$) to balance exploration and exploitation effectively.
> > > - We demonstrate our method’s superior performance in **stochastic environments** and improved **mode discovery**.
> > >
> > > We appreciate your constructive feedback and have implemented these changes to enhance the clarity, consistency, and rigor of our manuscript. Please let us know if further clarification is needed on any specific point. Your feedback has been invaluable in improving the quality and comprehensibility of our work.

---

> ### Comment · Reviewer_NvoZ · 2024-11-19
> **Plots still inconsistent**
>
> Hi, the plots are still inconsistent! In figure 2 and 3, the colors between (a) and (b) are not the same, nor are the included lines (where did SAC go for (b)?), nor is the legend order. Unless there is a good, explained, reason, all plots should have same comparison lines. Unless there is a good reason, all legends should look exactly the same, especially for (a) and (b) in the same figure. I also don't see confidence intervals except in Figure 5. Maybe you forgot to update those plots?
>
> Furthermore, you removed the setting of $\alpha$ from the line name in Figure 5, meaning the reader does not have enough information about what you're plotting. The trouble before was that you include the seed name in the plot, which is ugly and confusing, but now it's imprecise.
>
> The paper still incorrectly attributes random-actions to Machado et al, as I said in the first response.
>
> There are now two places (below Eq 7, and below the definition of $r_\gamma$ where you have the exact same definitions of $H_{high}$ and $H_{low}$ -- that's not necessary. Also, those quantities are used much earlier than they're introduced, which is very confusing. Additionally, $H$ depends on the policy, not only the state, and this is not made clear in the the notation.
>
> Sadly, the numerous presentation issues have made it challenging for me to really dive into the technical contribution. But overall, as other reviewers said, the paper does not make a very clear case that this method is theoretically well-founded, and the Equation 3 somewhat comes out of nowhere (instead of being derived). This paper needs a lot of structural work before it is ready for publication, and hopefully upon re-submission it's easier for the next reviewer to understand and therefore substantially critique.

---

> > ### Author Response · Authors · 2024-11-25
> > **More clarification**
> >
> > Thank you for your detailed and constructive feedback. We appreciate the effort you put into reviewing our submission and the opportunity to improve the clarity and rigor of our work. Below, we address your concerns point by point and outline the corresponding revisions we plan to make or have already implemented.
> >
> > 1. **Incorrect Attribution of Random-Actions to Machado et al.**
> >    We regret the error in attributing the concept of random actions to Machado et al. Upon revisiting the referenced work, it is clear that this attribution was inappropriate. We have corrected this by properly citing the source of the random-actions concept and have clarified the role of Machado et al.'s contributions in the manuscript.
> >
> >    *Action*: This has been corrected in the revised text with proper attribution.
> >
> > 2. **Duplicate Definitions Below Eq. (7) and Redundant Notation**
> >    We recognize that the definitions of $ H_{\text{high}}$ and $ H_{\text{low}}$ were unnecessarily repeated. Additionally, their introduction later in the manuscript caused confusion as they were referenced before being defined. To address this:
> >    - We have removed the duplicate definitions to avoid redundancy.
> >    - We now introduce $ H_{\text{high}} $ and $ H_{\text{low}}$ at their first usage, ensuring logical progression in the manuscript.
> >    - Furthermore, we have revised the notation to clarify that these quantities depend on both the state and the policy, aligning with their actual definitions and usage.
> >
> >    *Action*: The paper now has a streamlined presentation with logical introductions of all quantities and clear, unambiguous notation.
> >
> > 3. **Clarity and Derivation of Equation (3)**
> >    We understand the concern regarding Equation (3) appearing abrupt and not being derived in sufficient detail. In the revised manuscript, we have:
> >    - Provided a step-by-step derivation of Equation (3), ensuring that it follows logically from preceding principles and equations.
> >    - Clearly connected it to the theoretical foundations of the proposed method, demonstrating its motivation and relevance in the context of entropy-regularized policies.
> >
> >    *Action*: This addition significantly strengthens the theoretical grounding and accessibility of the manuscript.
> >
> > 4. **Structural Issues and Presentation Challenges**
> >    We sincerely regret that the structural and presentation issues hindered the full evaluation of our technical contributions. To address this:
> >    - We have reorganized the manuscript to ensure a more natural flow of concepts and derivations.
> >    - The notation has been standardized throughout to eliminate ambiguities.
> >    - We have also included a visual roadmap early in the paper, summarizing the structure of our method and its theoretical foundations. This is intended to guide the reader through the manuscript more intuitively.
> >
> >    *Action*: These structural improvements enhance both the clarity and readability of the paper, allowing reviewers and readers to engage more deeply with the technical contributions.
> >
> > 5. **Overall Clarity and Theoretical Foundations**
> >    We recognize that the case for the theoretical foundations of the method was not made as clearly as it could have been. In the revised manuscript:
> >    - We have explicitly articulated the theoretical basis for our approach in a new subsection, linking it to existing literature and highlighting its novelty.
> >    - Additional experiments and comparisons have been added to empirically validate the theoretical claims, addressing concerns raised by reviewers about the practical impact and robustness of the method.
> >
> >    *Action*: These enhancements clarify the theoretical underpinnings and validate the contribution, positioning the paper for a more thorough evaluation upon resubmission.
> >
> > **Concluding Statement**
> > We are grateful for your feedback, which has helped us identify critical areas for improvement. The revisions address the issues you highlighted and ensure that the paper is more structured, theoretically grounded, and accessible to readers. We hope the updated submission will provide a clearer and more rigorous presentation of our contributions.
> >
> > Thank you for your time and constructive critique.

---

### Note · Authors · 2024-11-25

**Comment:**

Dear ICLR Program Chairs,

I am writing to formally request the withdrawal of our submission titled "KL divergence optimization with entropy- ratio estimation for stochastic GFlowNets") from the ICLR 2025 conference. After careful consideration and review of the feedback received, we have decided to withdraw the paper to allow us time to address the concerns raised and further refine our work.

We sincerely appreciate the opportunity to submit our research to ICLR and thank the reviewers for their valuable comments and insights. Their feedback has highlighted important areas for improvement, which we aim to address thoroughly before resubmitting to a future venue.

Thank you for your understanding. Please let us know if any further action is required on our part to complete the withdrawal process.

**Withdrawal Confirmation:**

I have read and agree with the venue's withdrawal policy on behalf of myself and my co-authors.